# The Application of Creatine Supplementation in Medical Rehabilitation

**DOI:** 10.3390/nu13061825

**Published:** 2021-05-27

**Authors:** Kylie K. Harmon, Jeffrey R. Stout, David H. Fukuda, Patrick S. Pabian, Eric S. Rawson, Matt S. Stock

**Affiliations:** 1Neuromuscular Plasticity Laboratory, Institute of Exercise Physiology and Rehabilitation Science, School of Kinesiology and Physical Therapy, University of Central Florida, Orlando, FL 32816, USA; kylie.harmon@ucf.edu; 2Physiology of Work and Exercise Response (POWER) Laboratory, Institute of Exercise Physiology and Rehabilitation Science, School of Kinesiology and Physical Therapy, University of Central Florida, Orlando, FL 32816, USA; Jeffrey.Stout@ucf.edu (J.R.S.); david.fukuda@ucf.edu (D.H.F.); 3Musculoskeletal Research Laboratory, Institute of Exercise Physiology and Rehabilitation Science, School of Kinesiology and Physical Therapy, University of Central Florida, Orlando, FL 32816, USA; Patrick.pabian@ucf.edu; 4Department of Health, Nutrition, and Exercise Science, Messiah University, Mechanicsburg, PA 17055, USA; erawson@messiah.edu

**Keywords:** supplements, muscle damage, recovery, immobilization, atrophy, muscular dystrophy, amyotrophic lateral sclerosis, Parkinson’s Disease, cardiopulmonary disease, mitochondrial cytopathy

## Abstract

Numerous health conditions affecting the musculoskeletal, cardiopulmonary, and nervous systems can result in physical dysfunction, impaired performance, muscle weakness, and disuse-induced atrophy. Due to its well-documented anabolic potential, creatine monohydrate has been investigated as a supplemental agent to mitigate the loss of muscle mass and function in a variety of acute and chronic conditions. A review of the literature was conducted to assess the current state of knowledge regarding the effects of creatine supplementation on rehabilitation from immobilization and injury, neurodegenerative diseases, cardiopulmonary disease, and other muscular disorders. Several of the findings are encouraging, showcasing creatine’s potential efficacy as a supplemental agent via preservation of muscle mass, strength, and physical function; however, the results are not consistent. For multiple diseases, only a few creatine studies with small sample sizes have been published, making it difficult to draw definitive conclusions. Rationale for discordant findings is further complicated by differences in disease pathologies, intervention protocols, creatine dosing and duration, and patient population. While creatine supplementation demonstrates promise as a therapeutic aid, more research is needed to fill gaps in knowledge within medical rehabilitation.

## 1. Introduction

Physical rehabilitation is beneficial for everyone, including children, adults, and older people with a wide range of health conditions. The goal of physical rehabilitation is to enhance, restore, or maximize functional ability to promote health and optimize quality of life. Globally, an estimated 2.4 billion people live with a health condition that would benefit from rehabilitation [1].

Numerous acute and chronic conditions can result in dysfunction and reduced health due to physical manifestations directly related to the disorder, such as orthopedic injury with subsequent disuse and post-surgical immobilization. Dysfunction also leads to inactivity, altered movement, or disuse, resulting in muscle mass loss. These symptoms can therefore also occur in association with cardiovascular disease, neurodegenerative conditions, or other disease conditions such as myopathies [2,3]. Further, in older populations, significant loss of muscle and function has been associated with increased mortality risk [4]; thus, interventions focused on regaining muscle loss must be prioritized.

Muscle atrophy is related to metabolic and physiological changes, including decreased sensorimotor excitability, contractile ability of muscle fibers, increased protein degradation, and depletion of energy stores, which may promote fatigue, reduce functional capacity and hinder activities of daily living [5,6,7]. More recently, there has been an increased appreciation for nutritional and dietary supplement interventions, both independently and combined with exercise, to maintain or enhance clinically essential outcomes and improve rehabilitation-related adaptations in patients experiencing substantial muscle and strength reduction [8,9]. Evidence suggests that one nutritional intervention in particular, creatine monohydrate supplementation, may effectively attenuate and possibly enhance rehabilitation [7,8,10].

Even though creatine was detected in skeletal muscle 189 years ago, irrespective of its immense success as an ergogenic aid since the 1990s, much remains unknown about its biological capabilities. This is especially true for its potential of use in accordance with health and medical conditions. Chemically known as a non-protein nitrogen compound [11], approximately 95 percent of the body’s creatine is stored within skeletal muscle. Moreover, small amounts of creatine are also found in the brain and testes [12,13]. Approximately two-thirds of the creatine found in skeletal muscle is stored as phosphocreatine, while the remaining quantity is stored as free creatine [13]. The standard concentration of total creatine in skeletal muscle is approximately 120 mmol·kg^−1^ (dry mass) [14,15], whereas the upper limit appears to be approximately 150–160 mmol·kg^−1^ [16,17,18,19]. It has been suggested that creatine’s physiological and biochemical effects are mostly related to the functions of creatine kinase and phosphocreatine (i.e., CK/PCr system) to support the maintenance of cellular energy [17,18,19]. The most commonly accepted theory explaining the beneficial effects of creatine supplementation on muscle performance is the energetic theory, that the increase in skeletal muscle concentrations of creatine and phosphocreatine contribute to an increase in exercise intensity and volume, thus providing a more substantial stimulus and leading to the significantly greater adaptation to training [13].

Creatine has gained popularity as a dietary supplement because of its fundamental role in human health and physical performance. Its metabolic contributions are many and wide ranging, and it is one of the most frequently used and effective supplements available for health and performance [20]. Various research studies over the last several decades have shown that creatine supplementation has numerous positive effects on skeletal muscle structure and metabolism, including improving training volume and muscle mass (growth/hypertrophy) in conjunction with exercise [20]. Therefore, the potential of creatine to augment training-induced muscle hypertrophy is not only relevant in the context of elite sports performance but also in the context of rehabilitation of various conditions due to its ability to prevent muscle atrophy [21]. While its role as an ergogenic agent in sports is well supported by several studies [12], there is growing research interest from the medical field regarding the potential for creatine to serve as a therapeutic agent in a wide range of conditions [22,23]. Figure 1 provides an overview of the therapeutic efficacy of creatine supplementation in a variety of conditions examined within this review. Interestingly, Corn and colleagues [24] have suggested that creatine may be beneficial as an adjunct to traditional physical therapy interventions. Therefore, the purpose of this review is to examine the literature concerning the effectiveness of creatine monohydrate supplementation as a potential strategy to enhance the rehabilitation of muscle and function in physical inactivity (disuse) and disease-related conditions.

## 2. Methods

A comprehensive review of the scientific and medical literature was conducted to assess the state of the science on creatine supplementation in medical rehabilitation. Specifically, the National Institutes for Health National Library of Medicine PubMed.gov and Google Scholar search engines were used to identify publications focused on the utility of creatine supplementation during rehabilitation due to immobilization, muscle disuse atrophy, neurodegenerative disorders, cardiopulmonary disease, and mitochondrial cytopathies.

## 3. Creatine Supplementation, Muscle Damage, and Recovery from Stressful Exercise

There have been anecdotal reports in the media that creatine supplementation is associated with increased muscle dysfunction, including increased incidence of cramps, injuries, and rhabdomyolysis [25,26,27,28,29]. However, this is not supported by the literature. The results of several studies show no increase or decrease in muscle cramps, tightness, strains, total injuries, missed practices, or players lost for the season in collegiate athletes ingesting creatine supplements [30,31,32]. Interestingly, even though anecdotal reports claim that creatine supplements exacerbate muscle dysfunction, several research teams have investigated creatine supplementation as a means to enhance recovery from intense exercise by reducing exercise-induced muscle damage [33,34,35]. Therefore, the potential benefit of creatine supplementation to reduce muscle damage and enhance recovery from stressful exercise is worthy of further investigation.

### 3.1. Mechanisms of Benefit

During stressful and particularly unaccustomed exercise, damage to muscle fibers can occur. In the days following exercise, muscle force production and range of motion decrease, while muscle soreness, swelling, muscle serum proteins, and inflammatory compounds increase [36,37]. However, creatine supplementation may offer several potentially beneficial effects in terms of recovery from damaging exercise. Increasing muscle creatine through supplementation may reduce muscle membrane fluidity and increase membrane stiffness, which could decrease damage from exercise [21,33,34,35]. Additionally, increased muscle creatine levels may help maintain calcium homeostasis, mitigate inflammation, and decrease free radical-induced damage following damaging exercise [21,33,34,35]. Beyond attenuating muscle damage, increased muscle creatine due to creatine supplementation alters the intramuscular milieu, which subsequently causes several changes beneficial to the adaptive response to resistance exercise. For example, creatine supplementation results in increased growth factor expression (e.g., myogenin, MRF-4, insulin-like growth factor I and II [IGF-I and IGF-II]) [38,39,40,41], increased satellite cell number and myonuclei concentration [42], and expression of multiple genes associated with adaptive processes related to exercise (e.g., osmosensing, cytoskeleton, remodeling, GLUT-4 translocation, glycogen and protein synthesis, satellite cell proliferation, and differentiation, DNA replication and repair, mRNA processing and transcription, and cell survival) [43]. It is possible that the increase in intramuscular water associated with creatine supplementation modulates some of the effects, as this is known to inhibit protein breakdown and RNA degradation and stimulate protein, DNA, RNA, and glycogen synthesis [33,44,45]. Collectively, these data support that creatine is not only a performance enhancing nutrient that provides greater fuel availability prior to intense exercise, but also an adaptive nutrient which, when supplemented, augments the adaptive response to training. 

### 3.2. Specific Effects on Muscle Damage

Our review of the literature resulted in the examination of 16 clinical trials that describe the effects of creatine ingestion on markers of muscle damage in hard training individuals (e.g., resistance exercise) or those subjected to an unaccustomed exercise challenge (e.g., high-force eccentric exercise) [46,47,48,49,50,51,52,53,54,55,56,57,58,59,60,61]. The methods used in these investigations are discrepant, including differences in supplementation dosing and duration, participant training status, and exercise challenge (e.g., resistance, high-force eccentric, and endurance) [25,34,35]. Across these studies, improvements are noted in established markers of exercise-induced muscle damage, including reduced post-exercise levels of muscle serum proteins (creatine kinase, lactate dehydrogenase); reduced inflammatory compounds (prostaglandin-E_2_, tumor necrosis factor-α, interferon-α, interleuikin 1-β), reduced oxidative stress markers (glutathione peroxidase, thiobarbituric acid reaction substances), increased recovery of strength, and reduced delayed onset muscle soreness. However, consistent improvements were not found across studies or across the same variables. In their systematic and meta-analytic review of creatine supplementation and muscle damage, Northeast and Clifford [35] concluded that creatine supplementation had little practical value as a recovery aid. However, they pointed out that there were less than 10 eligible studies for each outcome and time point, limiting statistical power and contributing to high heterogeneity for outcomes. Two important conclusions are worthy of consideration. First, as several studies have shown a protective effect of creatine supplementation on exercise-induced muscle damage, this area requires further exploration. As small sample size prevented previous reviews from conducting sub-analyses [35], we cannot know if creatine supplementation might benefit one population (e.g., trained vs. untrained) or be more protective in a specific type of exercise (e.g., endurance vs. eccentric-resistance). Secondly, despite lingering myths and anecdote, there are no data to support that creatine supplementation increases muscle damage following severe and/or unaccustomed exercise, which in some studies was quite severe (e.g., 50 maximal eccentric contractions [55]; Ironman triathlon [48]).

## 4. Rehabilitation and Creatine Use for Clinical Conditions

### 4.1. Immobilization and Disuse-Induced Atrophy

Skeletal muscle disuse has been repeatedly demonstrated as having deleterious effects on a variety of physiological parameters. Even short periods of disuse have caused observable decrements in muscle cross-sectional area (CSA) [62], reduced force production capabilities [63], increased muscle protein breakdown [64], and altered neuromuscular function [65]. The decrements that occur during disuse can lead to longer recovery periods, a greater chance of injury recurrence, and a decline in metabolic health [66,67,68,69,70]. Because of its high clinical relevance, joint immobilization has become a frequently employed experimental model to examine physiological changes in skeletal muscle as a consequence of inactivity and disuse.

Due to the well-documented ability of creatine to potentiate the anabolic effects of resistance training [71], the potential of creatine supplementation to mitigate the effects of disuse-induced maladaptation is of exceptional interest [72,73]. Indeed, seven recent studies have examined the effects of creatine supplementation during immobilization, with several having demonstrated promising results. Using a single-blind, cross-over design, Johnston et al. [74] observed the effects of creatine supplementation on the preservation of muscle mass, strength, and endurance after a 7 day immobilization period of the upper limb in creatine-naïve men. Compared with an isocaloric placebo, 20 g·day^−1^ (four doses of 5 g each) of creatine supplementation during the immobilization period better maintained lean tissue mass, elbow flexor strength, and endurance, and elbow extensor strength and endurance. Similarly, in a double-blind, placebo-controlled trial involving 2 weeks of lower-limb cast immobilization during which participants ingested either creatine or a placebo, Eijinde et al. [75] observed that 20 g·day^−1^ (four doses of 5 g each) of creatine supplementation offset the observed decreases in muscle GLUT4 protein content that occurred in the placebo group. Following a subsequent 10-week rehabilitation training period during which creatine supplementation was reduced (15 g·day^−1^ during the first 3 weeks and 5 g·day^−1^ for the remaining 7 weeks), muscle GLUT4 protein content was increased by ~40% in the creatine group while returning the baseline levels in the placebo group. Given the role of muscle GLUT4 protein in providing a mechanism for glucose to enter the cell, these results suggest that creatine supplementation may promote favorable changes in glucoregulation during immobilization.

The efficacy of creatine supplementation during periods of inactivity has also been observed in animal models. Hindlimb suspension, often used as a surrogate for cast immobilization, causes observable muscle atrophy and strength loss in rodent models [76]. Aoki et al. [77] investigated the effect of creatine supplementation (5 g·kg^−1^·body weight·day^−1^) both prior to and during a period of hindlimb suspension in rats. The animals underwent 7 days of hindlimb immobilization in combination with three supplemental interventions: creatine during the immobilization period, creatine for 7 days prior to and during immobilization (i.e., 14 days of creatine supplementation), and a control condition. Regardless of the treatment group, immobilization induced a decrease in muscle weight. However, creatine supplementation prior to and during immobilization appeared to mitigate muscle mass loss. This is an intriguing finding, as a period of creatine loading is a frequent practice in athletic populations [20] and could easily be incorporated into pre-treatment or preoperative protocols in patients undergoing periods of planned immobilization to attenuate muscle atrophy.

While the aforementioned findings are promising, other works provide conflicting results. Marzuca-Nassr et al. [78] observed inconclusive effects of short-term creatine supplementation during hindlimb immobilization in rats. Throughout 5 days of hindlimb immobilization, adult rats received daily creatine supplementation (5 g·kg^−1^ body weight day^−1^) or a placebo. Despite positive changes in protein metabolism and a slight attenuation of total muscle loss in the creatine group, creatine supplementation did not prevent muscle fiber atrophy or strength loss. Three double-blind, placebo-controlled trials in humans have also demonstrated inconsistent findings of creatine supplementation throughout immobilization. Fransen et al. [79] observed no effect of 20 g·day^−1^ creatine supplementation on maintenance of work or power production following 7 days of wrist cast immobilization. Similarly, throughout 7 days of lower-limb cast immobilization during which participants received a loading dose of creatine (20 g·day^−1^ for 5 days prior to immobilization) followed by a maintenance dose (5 g·day^1^) during the immobilization period, Backx et al. [80] found no effect of creatine supplementation on preservation of quadriceps CSA or knee-extension strength. While the loading dose of creatine was successful in increasing muscle creatine content, this was ineffective in preservation of strength or muscle mass. These results persisted upon the removal of creatine non-responders (i.e., participants who did not show an increase in muscle total creatine content exceeding 10 mmol·kg^−1^) from the analyses and throughout a subsequent 7 day recovery period. Similarly, during 2 weeks of lower-limb cast immobilization in healthy young men, Hespel et al. [7] found no effect of 20 g·day^−1^ of daily creatine supplementation in preserving quadriceps CSA or knee-extension power. However, throughout a subsequent 10-week program of rehabilitation training during which creatine supplementation was reduced (15 g·day^−1^ the initial 3 weeks and 5 g·day^−1^ during the remaining 7 weeks), quadriceps CSA and knee-extension power recovered at a faster rate in participants supplementing with creatine versus placebo. Further, myogenic protein expression was altered in the creatine group during rehabilitation. In particular, MRF-4 expression was significantly increased in the creatine group during rehabilitation and strongly correlated to changes in muscle fiber diameter. While the anabolic potential of creatine in combination with resistance training is well documented, its efficacy in providing strategies to increase the rate of functional recovery during rehabilitation is a significant finding.

While the specific mechanisms involved in the maintenance of muscle mass during immobilization are not fully known, based on the literature, it is likely that creatine supplementation may have a protective effect on skeletal muscle at least partially due to alterations in muscle protein expression [7,41,78] and satellite cell activity [42]. Given the prolonged recovery time often observed after clinical immobilization [67], interventions to enhance the rehabilitation process are sorely needed.

#### 4.1.1. Post-Operative Orthopedic Recovery

Despite the promising results observed during rehabilitation from immobilization [7], creatine supplementation does not appear to confer a beneficial effect during post-operative rehabilitation. In a randomized, double-blind, placebo-controlled trial, Tyler et al. [81] observed the effect of creatine supplementation on recovery of muscle strength following anterior cruciate ligament (ACL) reconstruction. Creatine supplementation began with a loading dose of 20 g·day^−1^ the day following reconstructive surgery. This dosage was decreased to 5 g·day^−1^ at day 7 when formal rehabilitation began and continued for 12 weeks. Quadriceps and hamstrings strength and power were unaffected by creatine supplementation. Utilizing a similar randomized, double-blind, placebo-controlled design, Roy et al. [82] observed no effect of creatine supplementation on functional recovery after total knee arthroplasty. In contrast to the work of Tyler et al. [81], patients in this trial received creatine pre-surgery (10 g·day^−1^ for 10 days) in addition to post-surgery (5 g·day^−1^ for 30 days). Despite the addition of a pre-surgery loading dose, creatine supplementation had no effect on preserving muscle strength or enhancing recovery, as measured by quadriceps, ankle, and handgrip strength, as well as timed walks and step climbs.

#### 4.1.2. Acute Injury

Like periods of immobilization, muscle and nerve damage can result in muscle atrophy as a consequence of impaired function, mobility, and physical inactivity. While medical and surgical interventions are available, therapies aimed at restoring function after severe damage can still result in imperfect clinical outcomes [83]. Given the extensive recovery time from muscle and nerve damage, agents that can enhance the healing process are critically needed.

Although the literature is limited, the effects of creatine supplementation during recovery from nerve damage have demonstrated promising results. Özkan et al. [83] observed a positive effect of creatine supplementation on reinnervation of denervated muscle in the rodent model. After severing the sciatic nerve, adult rats were fed a creatine-enhanced or normal diet during subsequent recovery, with subgroups of animals undergoing surgical nerve repair and others receiving no neural anastomosis. In both those with and without surgical nerve repair, creatine supplementation significantly improved functional recovery as measured by walking analyses, pinch strength, limb circumference, and toe contracture. To the best of our knowledge, this has yet to be replicated in humans; however, this is a noteworthy finding as it indicates that supplementation with creatine may enhance recovery after denervation.

Despite the potential efficacy of creatine supplementation in recovery from nerve damage, similar effects were not observed during recovery from muscle damage in animal models [84]. After injecting rat soleus muscle with notexin to cause myotoxin-induced muscle degeneration, rats were fed a creatine-enhanced or normal diet and observed for 42 days post-injury. While creatine supplementation was able to ensure faster recovery of total muscle creatine and phosphocreatine content, it did not influence regeneration of muscle mass to pre-injury levels and had no effects on the time course of recovery. However, physical activity, which has previously been demonstrated to have a strong effect on the efficacy of both creatine supplementation and muscle recovery [85], was not manipulated. Future research should examine the effects of combined creatine supplementation and physical activity on muscle regenerative capacity.

#### 4.1.3. Spinal Cord Injury

Individuals with spinal cord injury (SCI) frequently suffer from extreme deconditioning and muscle strength impairments due to the nature of their injuries [86]. Patients with SCI often experience diminished upper body strength and reduced work capacity which can lead to inactivity, further exacerbating these problems [87,88,89]. As such, patients with SCI would benefit greatly from increased levels of muscular strength, power, and endurance to improve movement for normal activities in their daily lives. Optimizing movement for activities such as transferring body weight, propelling a wheelchair, or navigating community obstacles would allow for more independence, increased activity, and improved health and quality of life.

Creatine supplementation has been investigated in individuals with SCI due to its potential as an anabolic substance. Although the literature is limited, the findings are promising. In a randomized, double-blind placebo-controlled cross-over trial of patients with cervical-level SCI, 20 g·day^−1^ of supplemental creatine enhanced upper extremity work capacity [90]. Patients with complete cervical-level SCI were supplemented with creatine or a placebo and performed incremental work capacity tests before and after 7 days of treatment. After creatine supplementation, participants had significantly improved VO_2_, VCO_2_, and ventilatory threshold as measured during incremental upper arm ergometry. These findings are further strengthened by the work of Amorim et al. [91], who, supplemented SCI patients with 3 g·day^−1^ creatine, vitamin D, or placebo throughout a progressive 8 week resistance training program in a randomized, double-blind, placebo-controlled fashion. In the creatine-supplemented group, upper body strength and corrected arm CSA significantly improved. This is particularly notable given the considerably lower daily dose of creatine compared to other studies. The findings from these studies indicate that creatine supplementation enhances the effects of exercise training in individuals with SCI. Similar results have also been demonstrated in rats with SCI [92]. Rats fed a creatine-enhanced diet for 4 weeks prior to surgical SCI demonstrated significantly better post-traumatic locomotor capacity than control rats. Creatine also appeared to reduce the effects of secondary neurotrauma as observed via reduction in lesion site scar tissue.

#### 4.1.4. Arthritic Diseases

Creatine supplementation may also confer a beneficial effect on physical function in those with arthritic diseases [93]. In a randomized, double-blind, placebo-controlled fashion, post-menopausal women with knee osteoarthritis demonstrated significantly improved physical function and lower-limb lean mass when supplemented daily with both a loading dose (20 g·day^−1^ for 7 days) and maintenance dose (5 g·day^−1^ for 11 weeks) of creatine throughout a 12 week resistance training program [94]. Similar findings have been observed in those with rheumatoid arthritis [95]. Administration of oral creatine over a 3-week period (20 g·day^−1^ for 5 days followed by 2 g·day^−1^ for 16 days) improved muscle strength in patients with rheumatoid arthritis, despite no associated training protocol over the course of the intervention. It should be noted, however, that although strength did improve, total skeletal muscle creatine content did not. The authors concluded no clear clinical benefit of creatine supplementation, as functional ability and disease activity did not change. Still, positive associations have recently been observed between strength and walking self-efficacy, pain reduction, and improved function in osteoarthritis, further demonstrating the importance of maintenance of muscle strength in patients with arthritic diseases [96].

#### 4.1.5. Rationale for Discrepant Findings throughout the Immobilization Literature

There are a variety of methodological and physiological reasons for the inconsistent observations of creatine supplementation during immobilization. Although it is difficult to compare results across studies given differing protocols, durations, and dosing, it has been suggested that the length of the immobilization protocol and muscle group involved impact the atrophic response and functional impairments [97]. It has previously been demonstrated that lower body muscle groups are more negatively impacted by disuse than upper body musculature [98,99]. It is therefore possible that creatine supplementation may have a differing effect on upper versus lower body musculature.

Further, the overall duration of creatine supplementation may significantly impact potential findings. While short-term creatine supplementation has been shown to increase muscle creatine content, longer-term creatine supplementation is associated with enhanced muscular hypertrophy and strength [100]. Given the acute nature of the bulk of the literature observing the effects of creatine on disuse-induced adaptations, it is possible that interventions utilizing creatine supplementation to counteract the effects of disuse would demonstrate greater efficacy from longer intervention protocols. When considering the positive effects of creatine loading in mitigating disuse-induced atrophy [77] as well as the increased hypertrophy [7] and strength [94] observed during longer protocols of creatine supplementation, increasing the duration of the supplementation seems promising. Further, several aforementioned studies with favorable results have incorporated exercise interventions in combination with creatine supplementation. As exercise has been demonstrated to enhance the uptake of creatine [14], the role of exercise in the efficacy of creatine supplementation is of critical importance and should be incorporated when mobility and function allow. Additionally, differences in the presence of a loading phase and maintenance dose alter the overall exposure to creatine supplementation. Loading doses of creatine are routinely administered as the greatest uptake of creatine in the muscle appears to occur in the initial phases of a heightened dose [14].

Still, it is possible that other factors are responsible for the discrepant findings in the literature. Neural impairments, such as decreased voluntary activation, have also been observed throughout immobilization [101] and can impact both muscular strength and endurance [102]. Significant alterations in neuromuscular properties, such as reduced voluntary activation [103,104], decreased H-reflex [105], and changes in motor cortex excitability [103,105] have been observed to occur with disuse, with neural factors explaining nearly 50% of initial strength loss [104,105]. If injury precedes disuse, alterations may occur in sensory stimuli, receptor activity, and signal transduction from the injured area to the central nervous system, further exacerbating functional impairments [106].

Finally, the effects of anabolic resistance, which have been repeatedly observed during periods of immobilization [107,108,109], must be taken into consideration. Anabolic resistance, the inability of an anabolic stimulus such as protein ingestion, hormonal response, or muscle contraction to stimulate muscle protein synthesis, is a hallmark characteristic of periods of inactivity or disuse [107,108,109]. Substantial inactivity or disuse decreases muscle protein synthesis with minimal change in muscle protein breakdown [110]. This can result in a dampened response to hyperaminoacidemia. This is significant, as one of the primary mechanisms postulated as responsible for creatine’s efficacy in improving muscular hypertrophy and performance is its ability to enhance protein stimulus via osmotic swelling [44,45,111]. However, if disuse-induced anabolic resistance is present, its effects may interfere with the protein synthesis that occurs with creatine supplementation, thereby inhibiting preservation of muscle mass and function. Similarly, the presence of satellite cells, which have been implicated in creatine’s efficacy, has been observed to decrease with disuse [112]. As satellite cells support regeneration of skeletal muscle following damage or atrophy, their role in recovery from immobilization or trauma is critical.

Promising results in immobilization studies may not be able to be extrapolated to clinical populations. While the bulk of the current disuse protocols in humans have been performed in healthy participants, patients undergoing immobilization associated with injury, surgery, or pain may experience accelerated atrophy and functional decrements. These phenomena may be partially responsible for the lack of creatine’s efficacy in post-operative observations. The physiological effects of illness or injury (e.g., inflammation, immune response) may exacerbate the negative changes in whole-body protein balance that occur during immobilization. Trauma, surgery, and pain may accelerate muscle atrophy during immobilization [113], ultimately leading to longer recovery times to return to baseline. Further, orthopedic issues severe enough to warrant surgery are likely to limit physical activity in advance of surgical intervention, leading to greater overall impairment. While creatine supplementation may offer enhanced outcomes from various conditions resulting in immobilization and injury, more work is needed to determine its overall efficacy in consideration of these factors.

### 4.2. Neurodegenerative Diseases

Neurodegenerative diseases are debilitating conditions caused by the progressive deterioration and eventual apoptosis of alpha motor neurons and skeletal muscle fibers. Given that these diseases are incurable, adjunctive therapies are needed to slow their progression and improve quality of life by controlling symptoms. As neurodegenerative diseases result in reduced physical activity, the theoretical basis of considering creatine supplementation in these patients is well justified, given its ability to improve muscle strength, mass, and endurance [20]. In other words, creatine supplementation may serve as a general countermeasure that delays the progression of physical impairments. However, neurodegenerative diseases adversely affect the distinct neurometabolic pathways and pathological characteristics that creatine may precisely target.

There are a variety of clinical design issues that have made studying the effects of creatine supplementation in patients with neurodegenerative diseases challenging. The most obvious challenge is that these conditions are uncommon, resulting in a shallow pool of patients that might be willing to enroll in clinical studies. In addition, the timeline and severity by which patients with neurodegenerative diseases experience declines in function can vary, resulting in heterogenous responses for both within and between groups. Rapid neurodegeneration and the need for a creatine washout period [100] make implementing a within-subjects design, which offer less error variance, more difficult. Patient care for neurodegenerative diseases is also incredibly complex and requires considerable resources. As such, this type of work is likely to be limited to hospital-based research centers and academic health science centers where patient care can be combined with carefully monitored clinical trials. Given these reasons, it is not surprising that prospective studies on creatine supplementation in patients with neurodegenerative diseases have featured small sample sizes and non-significant trends, highlighting the potential for type II statistical errors.

#### 4.2.1. Muscular Dystrophies

Skeletal muscle free creatine and phosphocreatine stores are significantly reduced in patients with certain myopathies and muscular dystrophies [114]. These observations have been linked to a lower creatine transporter protein content and impaired creatine uptake/release kinetics [115]. Our review of the literature shows that the potential therapeutic benefits of creatine supplementation have been studied in patients with Duchenne muscular dystrophy and Becker’s muscular dystrophy more than in other neurodegenerative diseases. These diseases are X chromosome-linked and caused by mutations with the dystrophin gene, with life expectancy in the 20′s for those with Duchenne muscular dystrophy. In most patients, shortened lifespan is mainly due to respiratory or cardiac failure [116]. It is interesting to note that the magnitude of creatine’s treatment effects has varied in patients with dystrophinopathies versus type 1 and type 2 myotonic dystrophy [117].

Voluntary muscle strength has been considered the primary outcome variable in most creatine studies in patients with muscular dystrophies. Three randomized, double-blind trials in boys with dystrophinopathies have reported noteworthy improvements. Louis et al. [118] assessed the effects of creatine supplementation in 12 boys with Duchenne muscular dystrophy and 3 boys (mean age = 11 years) with Becker’s muscular dystrophy. They utilized a randomized, double-blind, cross-over design, with a 2-month washout period in between. Following supplementation with creatine, a significant improvement in strength was observed and time to exhaustion during a submaximal contraction nearly doubled. Louis et al. [118] also reported a 25% increase in joint stiffness during the placebo period, but no change was observed while supplementing with creatine. Similar findings were reported by Tarnopolsky and colleagues [119], who tested the hypothesis that creatine supplementation would increase muscle strength and mass in boys (mean age = 10 years) with Duchenne muscular dystrophy while utilizing a randomized, double-blind, cross-over design. Following 4 months of creatine supplementation, significant improvements in grip strength of the dominant hand and fat-free mass were noted. Utilizing a 6 month, parallel-group design, Escolar et al. [120] noted strong trends for improvements in strength and other functional tasks that may have been mitigated by the unexpected maintenance of strength in the placebo group. Of these three studies, less variability was observed in the two cross-over trials [118,119] than in the parallel-group trial [120]. In contrast to studies on dystrophinopathies, creatine supplementation has not been shown to enhance strength or other measures of physical function in patients with type 1 myotonic dystrophy [121,122] or type 2 myotonic dystrophy/proximal myotonic myopathy [123]. This would suggest that the effects of creatine supplementation in patients with a specific type of muscular dystrophy may not be replicated or extrapolated to others. However, it should be noted that studies which have demonstrated benefits of creatine supplementation have been reported in children, whereas non-significant effects have been limited to adult patients. A considerable gap in knowledge is whether benefits of creatine supplementation in patients with certain myopathies and muscular dystrophies are age specific.

Beyond improvements in muscle strength, creatine supplementation in patients with muscular dystrophies may have two other benefits. First, creatine supplementation has generally resulted in improvements in self-reported activities of daily living or subjective improvement in function [122,123,124,125]. Work by Banerjee et al. [125], for example, demonstrated that 53.8% of parents whose children had been assigned to a creatine group reported “better” outcomes. Schneider-Gold et al. [123] reported that patients assigned to a creatine treatment group reported significant improvements in the subjective assessment of activity in daily life using a visual analogue scale. This was observed even though muscle strength did not improve [123]. Overall, patients with muscular dystrophies that supplement with creatine may experience subjective improvements in activity levels or function, regardless of changes in objective measures. Second, limited evidence suggests that creatine supplementation may enhance bone density in dystrophic children. Louis and colleagues [118] reported that in a subgroup of boys able to walk throughout their trial, creatine supplementation resulted in significant improvements in bone mineral density. The potential for preservation of bone mineral density is consistent with studies demonstrating a decrease in urinary N-telopeptide excretion (a marker of bone breakdown) [118,119]. Finally, no studies have reported side effects or adverse reactions to supplementing with creatine in patients with muscular dystrophies, many of whom have been young children. While more research is needed, the overall body of literature suggests that creatine supplementation shows promise as a safe and cost-effective means of maintaining or even enhancing muscle strength, functional performance, and bone density in patients with muscular dystrophies. Smaller improvements have been observed in lean mass [119,121,122] and with assessments of muscle strength that are not highly quantitative, such as manual muscle testing [123]. Responses to these variables seem to differ among specific types of dystrophies, highlighting the need for individualized treatments. It is important to further emphasize that reductions in strength and lean mass are hallmarks of these conditions. Small improvements or even maintenance over time may have real-world clinical value to patients and their families.

#### 4.2.2. Amyotrophic Lateral Sclerosis

Amyotrophic lateral sclerosis (ALS) is a neurodegenerative disease that results in the progressive loss of motor neurons that control voluntary muscle activity. It is the most common type of motor neuron disease and is always fatal [126]. Individuals diagnosed with ALS suffer from muscle weakness, atrophy, and difficulty with speech, among many other debilitating symptoms. With no cure available, most ALS medications are intended to minimize pain and fatigue [127].

Several authors have provided a rationale for why creatine supplementation would be a viable treatment option for individuals diagnosed with ALS [128,129]. Plausible explanations for why creatine might enhance various aspects of quality of life in patients with ALS include protection against neuron loss in the substantia nigra and motor cortex [130], as well as decreased oxidative stress [131] and mitochondrial dysfunction [132] commonly observed with this condition. Despite this initial enthusiasm and encouraging animal work [133], clinical trials in humans have reported disappointing results. Only three published trials utilizing a double-blind, randomized designed have evaluated the effects of creatine supplementation in ALS patients, and all three showed no benefits beyond placebo [134,135,136]. All three studies utilized doses of 5–10 g·day^−1^, measured muscle strength, and reported that creatine was well tolerated with no major side effects. Given that these studies included patients that were late in the disease process, it remains to be determined if use of creatine as a therapeutic adjunct early after diagnosis would result in better outcomes.

#### 4.2.3. Multiple Sclerosis

Multiple sclerosis (MS) is an immune-mediated disease caused by destruction or failure of myelin-producing cells, resulting in impaired nerve transmission. The usual onset of MS is between 20 and 50 years of age and it is twice as prevalent in women than men. The symptoms associated with MS are highly variable, but frequently include muscle weakness, difficulty with balance and vision, and fatigue. There is no known cure for MS, though physical therapy can help patients improve functionality.

There is a strong theoretical rationale as to why creatine supplementation may enhance outcomes in patients with MS [137]. For example, MS patients show compromised brain creatine metabolism [138], reduced cardiac phosphocreatine concentration [139], and elevated levels of creatine kinase in cerebrospinal fluid [140]. However, only two studies in patients with MS have evaluated the effects of creatine supplementation, with both reporting no improvements above placebo. Utilizing a randomized, double-blind, placebo-controlled trial, Lambert et al. [141] examined the effects of a loading dose (20 g·day^−1^ for 5 days) on isokinetic knee extension and flexion work. Vastus lateralis muscle biopsies were taken to measure intramuscular total creatine, phosphocreatine, and free creatine, and bioelectrical impedance analyses were used to examine body composition. The authors concluded that creatine supplementation had no influence on muscle creatine stores or high-intensity exercise capacity in patients with MS. These findings were supported by the work of Malin and colleagues [142], who, utilizing a 14 day, double-blind, cross-over trial with a three-week washout period, reported that creatine supplementation did not enhance knee joint power. Clearly, this area is ripe with opportunities for scientific discovery, but the results to date do not support the notion that creatine supplementation enhances physical function in MS.

#### 4.2.4. Parkinson’s Disease

Parkinson’s Disease (PD) is a progressive neurodegenerative disease that is characterized by resting tremors, rigidity, slowness, and problems with gait and balance. In addition to motor impairments, individuals with PD frequently report anxiety, apathy, and depression [143]. The main pathological characteristic of PD is cell death in the basal ganglia, leading to a reduction in brain dopamine levels. PD is typically diagnosed in adults around the ages of 50–60 years [144]. While a variety of pharmacological drugs are used to combat symptoms associated with PD, such as dopamine agonists, levodopa, and monoamine oxidase inhibitors, there is no cure, though the disease itself is not lethal [145]. As such, innovative approaches for managing symptoms and improving quality of life in those with PD are needed.

Given creatine’s well-documented ability to improve muscle function in healthy adults [17,20] some have speculated that individuals with PD may particularly benefit from supplementation [146]. Despite widespread interest, only five studies have compared creatine supplementation to placebo in a randomized, double-blind fashion in adults with PD [147,148,149,150,151]. Aside from one study by Bender, each of these five studies provided a dose of 10 g·day^−1^ to participants assigned to the creatine group. The most common outcome measures in these trials included the Unified Parkinson’s Disease Rating Scale and its constituent parts (e.g., Mental, Motor, and Activities of Daily Living). Studies by the National Institute of Neurological Disorders and Stroke (NINDS) [149] and Kieburtz [151] utilized the Schwab and England Scale, which assesses the difficulties patients have completing chores and daily activities. The findings from these studies have shown that the impact of creatine supplementation in individuals with PD may be small. Various aspects of these studies have demonstrated conflicting results. The NINDS study [149], for example, demonstrated no change in any of the Unified Parkinson’s Disease Rating Scale outcomes, but significant between-group differences for changes in the Schwab and England Scale. Collectively, the limited studies in this area have indicated that creatine supplementation may not be an effective therapeutic strategy for individuals with PD.

#### 4.2.5. Charcot–Marie–Tooth Disease

Charcot–Marie–Tooth Disease (CMT) is a group of inherited motor and sensory neuropathies that cause muscle atrophy and weakness in the hands and feet [152]. CMT is slowly progressive and incurable. While it is typically not fatal, most patients experience difficulty with muscle stiffness and gait due to foot drop and increased foot supination [153]. Therefore, patients with CMT experience a progressive decline in strength, physical activity, and function.

Only three studies have evaluated the effects of creatine supplementation in CMT patients. Doherty and colleagues [154] evaluated the potential benefits of one month of creatine supplementation in patients (mean age = 43 years) with CMT disease type 1 (*n* = 34) and type 2 (*n* = 5) using a double-blind, placebo-controlled, cross-over design. They reported no significant differences in visual analogue activities of daily living scales, body mass, estimated percentage body fat, or fat-free mass after creatine supplementation as compared with placebo. Shortly thereafter, the same research group published two studies which aimed to test the hypothesis that creatine supplementation would enhance strength and myosin heavy chain content in patients with CMT when combined with resistance training [155,156]. Utilizing a randomized, double-blind design, Chetlin et al. [155] evaluated 20 patients (mean age = 45 years) with CMT disease who completed a 12 week resistance exercise training program. The resistance training program was divided into three 4-week phases and was completed at home with adjustable wrist and ankle weights with a therapeutic squeeze ball. The authors reported that the resistance training program resulted in improvements in most outcomes, but there were not interactive effects for creatine supplementation. Given the large effects that resistance training elicits in novices [157], it is possible that any benefits of creatine were not detectable in the treatment group. Smith and colleagues [156] further analyzed these data to determine whether the combination of resistance exercise and creatine supplementation would increase the percentage of type I myosin heavy chain content composition in the vastus lateralis and whether myosin isoform changes would correlate with improved chair rise-time in patients with CMT. Their findings indicated that, when combined with resistance training, creatine supplementation resulted in a decline in type 1 myosin heavy chain content and an increase in type II myosin heavy chain content. Moreover, these changes correlated with an increase in chair rise performance. While speculative, the work of Smith et al. [156] points to a role for creatine supplementation altering skeletal muscle protein synthesis, activating satellite cells, and modifying myosin heavy chain isoform in patients with CMT. Given that only three studies have included patients with CMT [154,155,156], it is unclear whether creatine supplementation is an effective treatment approach for improving clinical outcomes. Given that there are so few therapeutic treatment options available for these patients, more work in this area is needed.

### 4.3. Cardiopulmonary Disease

Patients with cardiovascular impairments, including those with chronic obstructive pulmonary disease (COPD) and congestive heart failure, typically display similar comorbidities and etiological factors resulting in malnutrition and muscle dysfunction [158,159,160] that may benefit from creatine supplementation during the process of physical rehabilitation. From a broad perspective, creatine plays a role in both cardiac function [161] and vascular health [162] with declines in endogenous creatine a likely consequence of related dysfunction that may be augmented with exogenous supplementation. While the current review is focused on physical rehabilitation rather than the influence of creatine on cardiopulmonary diseases, in-depth analysis of these topics are presented by Balestrino [161] and Clarke et al. [162]. There is evidence to support performance benefits in the literature, with improved muscular strength and endurance, maximal aerobic power, and body composition in patients suffering from COPD [163] and heart failure [164,165,166] following creatine supplementation without exercise interventions. However, these findings are not consistent [167,168].

When examined in combination with cardiopulmonary rehabilitation, the potential clinical benefits of creatine supplementation for patients with COPD and congestive heart failure have been limited [169]. Cardiopulmonary rehabilitation programs typically feature two to three 60–90 min exercise sessions per week, completed over a 7- to 12-week period. Each session usually includes both aerobic and resistance training components, with an intended aim of progressive overload. Minimal improvements in physical performance, health-related quality of life, and body composition with the addition of creatine supplementation [168,170,171,172] have been observed. However, this may be due to the robust physiological stimulus and positive outcomes for these measures provided by structured exercise programming in these individuals.

Only one study [163] reported significant increases in fat-free mass, muscular strength/endurance, and health status in COPD patients receiving supplemental creatine versus placebo during pulmonary rehabilitation, but found similar changes in pulmonary function and whole-body exercise capacity between groups. Notably, the creatine group in this study had an average body mass index (BMI) of 23.2 ± 3.6 kg·m^−2,^ which is lower than those examined elsewhere (25–28 kg·m^−2^) and the closest to the cutoff of being classified as underweight (21 kg·m^−2^) with the potential of protein-energy malnutrition. Further, the study design featured the most comprehensive creatine loading protocol (5.7 g, 3× daily for two weeks) of the evaluated studies, which was completed prior to commencement of the pulmonary rehabilitation program and resulted in augmented body composition (body mass and fat-free mass) and muscle function (leg extension endurance and handgrip strength/endurance). In comparison, Deacon et al. [168] also utilized a creatine loading phase (~5.5 g, 4× daily for 5 days) in COPD patients before beginning a program of rehabilitative exercise in combination with a maintenance dose of creatine. Despite this, patients in both the creatine and placebo groups showed similar improvements after the intervention. However, it should be noted that the creatine group in this investigation had the highest BMI of the evaluated studies.

From a different perspective, Hemati et al. [173] reported improvements in markers of inflammation (interleukin-6 and C-reactive protein) as well as endothelial dysfunction (P-selectin and intercellular adhesion molecule-1) in heart failure patients undergoing an 8 week aerobic training program while supplementing with a 5 g·day^−1^ maintenance dose of creatine compared to a control group receiving no treatment. Recently, Ostojic [174] proposed that along with pulmonary rehabilitation, creatine supplementation be used as adjuvant therapy as nutritional support for those experiencing “long-haul” symptoms of COVID-19. Taken together with the findings in patients with cardiopulmonary impairments (with and without exercise), additional research and larger clinical trials [175] are still needed to evaluate the efficacy of creatine supplementation in these clinical populations.

### 4.4. Mitochondrial Cytopathies

Mitochondrial cytopathies are a heterogenous group of genetic disorders that adversely change the electron transport chain (ETC) and its function [176]. The most common mitochondrial cytopathy is mitochondrial encephalopathy, lactic acidosis, and stroke-like episodes (MELAS). Other types include chronic progressive external opthalmoplegia (CPEO) and Kearns–Sayre syndrome [176]. Patients with mitochondrial cytopathies show poor exercise tolerance, very low VO_2_ max values, and decreased ability to extract oxygen peripherally [177]. Interestingly, the decrease in aerobic power results in an up-regulation of the anaerobic pathways [177]. Further, mitochondria dysfunction results in an inability to meet various energy needs for proper function, notably in the nervous, cardiac, endocrine, and musculoskeletal systems [178,179]. This is especially evident within the muscular and nervous systems due to the high energy demands of muscle and nerve [179]. Since there is currently no known cure for these diseases, patients are managed and treated based upon their symptoms, which typically involve muscular weakness, ataxia, and intolerance to physical activity [180,181].

Nutritional interventions such as antioxidants have been recommended due to the high level of oxidative stress observed in patients with mitochondrial cytopathies, but to date have shown no efficacy. Interestingly, several investigators have shown a decrease in phosphocreatine/inorganic phosphate ratio and phosphocreatine concentration in patients with mitochondrial cytopathy. Thus, it appears that the functional loss of the ETC adversely alters phosphocreatine metabolism, which has been shown to delay recovery post-exercise [114,182,183]. Increasing the concentration of phosphocreatine, with creatine supplementation, in skeletal muscle has been shown to increase muscle function and accelerate recovery from exercise in healthy populations [184]. Furthermore, it has been suggested that creatine supplementation may also attenuate oxidative stress, thereby reducing free radical damage to the mitochondria [185]. Therefore, creatine supplementation would be a potentially beneficial strategy in this patient population.

Given the potential benefits of creatine supplementation in patients with mitochondrial cytopathies, several clinical studies have been conducted. Tarnopolsky and colleagues [186] administered supplemental creatine (10 g·day^−1^ for 14 days, followed by 4 g·day^−1^ for 7 days) in a double-blind fashion to primarily MELAS variant patients with severe mitochondrial cytopathy (VO_2_ max = 10.1 mL·kg^−1^·min^−1^). Creatine supplementation resulted in an 11% improvement in dorsiflexion strength endurance and a 19% improvement in handgrip strength. Further, although not statistically significant, there was a 0.6 kg increase in lean mass in the creatine group.

In contrast, in a randomized, placebo-controlled cross-over trial, Klopstock and colleagues [187] provided supplemental creatine (20 g·day^−1^ for 28 days) to 16 patients with primarily CPEO variant mitochondrial cytopathy. While the large dose was well tolerated, there were no observed significant effects on exercise performance measures or activities of daily living. Using a similar design, Kornblum et al. [188] provided patients with CPEO mitochondrial cytopathies with 150 mg·kg^−1^ of body weight per day of supplemental creatine for six weeks. In agreement with Klopstock et al. [187], creatine supplementation did not improve exercise performance measures. Further, intramuscular phosphocreatine/adenosine triphosphate ratio did not increase, and there was no effect on post-exercise phosphocreatine recovery.

There are several reasons for the conflicting results in examinations of creatine’s impact on mitochondrial cytopathies. Unlike the MELAS variant, CPEO patients typically have normal ETC enzyme activity and, more importantly, normal creatine and phosphocreatine concentrations in skeletal muscle. Due to the pathophysiology differences between MELAS and CPEO patients, responses to creatine supplementation may be different. For example, it is well known that patients with low skeletal muscle creatine and phosphocreatine levels respond to exercise performance measures to a greater extent with creatine supplementation [184]. Given that the CPEO patients in Kornblum et al. [188] had phosphocreatine levels that were not significantly lower than those in the control group may have resulted in a non-significant difference in response. Therefore, future studies examining the effect of creatine supplementation in this population need to identify the mitochondrial variant to determine the efficacy of the intervention. As indicated in a recent systematic review, sound research designs are challenging due to the heterogeneity in disorders and physical presentation [189]. More consistent study endpoints, design, and clinically relevant outcomes have yet to be determined, and should be considered with higher sample sizes [189].

While there is some evidence suggesting creatine supplementation combined with rehabilitation or treatments for other pathologies involving muscular dysfunction may be beneficial, there are very few clinical trials examining the effects of creatine for patients with mitochondrial cytopathies. Due to the vast array of possible physical presentations and limitations of patients with mitochondrial cytopathies, formal physical rehabilitation and physical therapy for patients should be individually-based on symptom presentation [181,190]. While treatment approaches for these patient populations demonstrate efficacy in the promotion of physical function, to the best of our knowledge no trials exist that integrate these physical interventions with the addition of creatine supplementation. Due to the potential efficacy as demonstrated individually, the combination of the two interventions may be promising for the promotion of function in patients with mitochondrial cytopathies.

## 5. Conclusions

Given the encouraging findings regarding the role of creatine supplementation throughout recovery from exercise, rehabilitation from immobilization or injury, and therapeutic support during various chronic conditions, creatine monohydrate demonstrates promise as a rehabilitative aid. Several notable findings have been reported. Based on the literature, creatine supplementation may:Support recovery from exercise by decreasing exercise-induced damage, supporting the adaptive response to exercise, and augmenting the physiological response to training. However, further research is needed regarding whether creatine supplementation confers a benefit to a specific population (e.g., trained vs. untrained) or type of exercise (e.g., endurance vs. resistance).Promote maintenance and mitigate the loss of muscle mass, muscular strength, and endurance, as well as promote healthy glucoregulation, during periods of immobilization. However, differing study protocols (e.g., muscle group involved, duration, creatine dosing) may impact the observed findings via differences in atrophic response, overall creatine exposure, alterations in neuromuscular mechanisms, and metabolic adaptations to disuse.Enhance recovery after nerve damage/denervation. While promising, these findings were observed in the rodent model and have not yet been replicated in humans.Improve physical function, lean mass, and muscular strength in populations with chronic arthritic diseases. However, the sample size of available trials is small, and more work is needed.Improve work capacity, strength, and lean mass in individuals with SCI. However, given the small number of available studies and limited sample size of participants, more work in this area is needed.Improve physical function, lean mass, muscular strength, bone density, and quality of life in patients with muscular dystrophy. Despite the promising findings in individuals with dystrophinopathies, similar findings have not been observed in patients with myotonic dystrophies, indicating that the effects of creatine supplementation are likely specific to dystrophy type.Confer a beneficial effect in patients with CMT disease via an increase in type II myosin heavy chain content, alterations in skeletal muscle protein synthesis, and activation of satellite cells. However, these findings are speculative, and more work is needed.Improve lean mass, muscular strength and endurance, and health status in COPD patients. However, observed responses may be complicated by the robust physiological response to exercise training in this population, and additional research and larger clinical trials are needed.Improve markers of inflammation and endothelial dysfunction in patients with heart failure. However, this has only been observed in a single trial, and more work in this area is needed.Improve muscular strength and endurance in certain mitochondrial cytopathies (MELAS variant). Due to the differences in pathophysiology of mitochondrial cytopathies, responses to creatine supplementation may differ between disorder, and should be further examined.

Notably, creatine supplementation appears safe and well tolerated in virtually all medical patient populations. Despite the positive impact of creatine supplementation in numerous clinical conditions, various gaps in the literature may prevent clinicians and medical professionals from strongly recommending that creatine be used to mitigate declines in physical function or for use while rehabilitating. More work in these areas is needed to gauge creatine’s role as a medical and rehabilitative aid.

## Figures and Tables

**Figure 1 nutrients-13-01825-f001:**
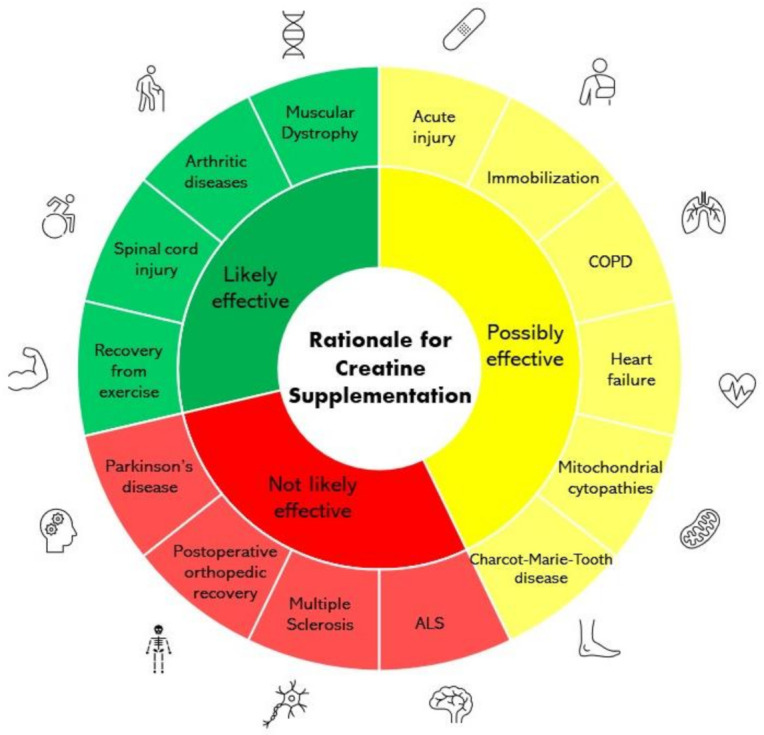
Rationale for examination of creatine supplementation as a rehabilitative aid. Abbreviations defined from clockwise: COPD = chronic obstructive pulmonary disease; ALS = amyotrophic lateral sclerosis.

## Data Availability

Not applicable.

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
