# Peer review of "The Application of Creatine Supplementation in Medical Rehabilitation"

_nutrients, 2021, doi:10.3390/nu13061825_

Round 1
Reviewer 1 Report
The manuscript of Harmon et al, conducted a review of the literature to ass the current state of knowledge regarding the effects of creatine supplementation on rehabilitation from immobilization and injury, neurodegenerative diseases, cardiopulmonary disease, and other muscular disorders. They found a rationale for discordant findings is further complicated by differences in disease pathologies, intervention protocols, creatine dosing and duration, and patient population, and propose more research to fill gaps in knowledge within medical rehabilitation.
INTRODUCTION
The introduction provides sufficient background information for readers to understand the research aim.
Motivations for this study are more than clear and the objectives are clearly defined at the Introduction, the argumentation in this part was concise.
Improve the quality of fig 1. Explain all the abbreviations in the figure.
METHODS
Explain database consulted, the keywords used, years, inclusion criteria…
RESULTS
The way to explain information and the division of it was good. It easy to follow the argument line and understand the information provided congratulation.
But to improve the quality of the paper takes into consideration:
In the 4.1 section - Explain the mechanism while creatine improve atrophy in Immobilization & disuse
In section 4.1.1. is not justified why creatine did not affect postoperative orthopaedic recovery
In section 4.1.3. explain the importance of creatine in spinal cord injury treatment and why it should be implemented
In 4.3. section. Creatinine has an impact on cardiopulmonary disease or just in body composition and then to the CD
In section 4.4. the mechanism of the creatin effect in mitochondrial cytopathies must be explain
CONCLUSION
According to the manuscript aims. The inclusion of key points allow to a better comprehension of the main results of the study
Author Response
The authors would like to thank the reviewer for their time, effort, and thoughtful comments. Our manuscript has undoubtedly been strengthened throughout the review process. Please see the attached document for specific responses to comments.

Reviewer 2 Report
The manuscript by Harmon et al. entitled "The application of creatine supplementation..." reviews the use of creatine as an efficacious therapeutic agent in the context of rehabilitation. There are a few points of clarification requested.
On line 78, the term "notoriety" is used, however, typically this term usually means being well-known for some bad quality. Is this the intended meaning regarding creatine and human health and physical performance?
It is stated on lines 108-110, that there are anecdotal reports of creatine and muscle dysfunction. Only one reference is provided to support this statement. Could additional citations be included for these reports or is this reference, viz., 24, a review of those reports?
Was Figure 1 adapted from any references or alternative resources or was it independently created by the authors?
In the conclusion section 5, each numbered example of eleven starts with the same phrase of "creatine supplementation," which is somewhat cumbersome. Please consider, "Based on the literature, several notable findings regarding creatine supplementation have been reported including the following" as one potential iteration. This would improve the flow of the list but is at the authors' discretion.
Author Response
The authors would like to thank the reviewer for their time, effort, and thoughtful comments. Our manuscript has undoubtedly been strengthened throughout the review process. Please see attached document for specific responses to comments.
